# Hypolipidemic, Hypoglycemic, and Ameliorative Effects of Boiled Parsley (*Petroselinum crispum*) and Mallow (*Corchorus olitorius*) Leaf Extracts in High-Fat Diet-Fed Rats

**DOI:** 10.3390/foods12234303

**Published:** 2023-11-28

**Authors:** Albandari A. Almutairi, Waheeba E. Ahmed, Raya Algonaiman, Raghad M. Alhomaid, Mona S. Almujaydil, Sami A. Althwab, Ard ElShifa M. Elhassan, Hassan Mirghani Mousa

**Affiliations:** 1Department of Food Science and Human Nutrition, College of Agriculture and Veterinary Medicine, Qassim University, Buraydah 51452, Saudi Arabia; 411200388@qu.edu.sa (A.A.A.); w.alfaki@qu.edu.sa (W.E.A.); r.alhomaid@qu.edu.sa (R.M.A.); m.almujaydil@qu.edu.sa (M.S.A.); thaoab@qu.edu.sa (S.A.A.); hasmousa@hotmail.com (H.M.M.); 2Department of Food Science and Technology, Alzaiem Alazhari University, Khartoum Bahri 13311, Sudan; 3Department of Chemistry, College of Science, Qassim University, Buraydah 51452, Saudi Arabia; ar.mohamed@qu.edu.sa

**Keywords:** antioxidants, plants, disease prevention, anti-obesity, obesity, diet, nutrition

## Abstract

Obesity is a major health concern associated with serious conditions such as type 2 diabetes and cardiovascular diseases. This study investigated the potential anti-obesity effects of heat-treated parsley and mallow extracts (PE and ME, respectively) in high-fat diet (HFD)-fed rats. The selected herbs underwent three heat treatments (boiling, blanching, and microwaving), and the most effective treatment was orally administered to the HFD rats for eight weeks. All three treatments effectively increased the total phenolic content (TPC) and antioxidant capacity of the herbs, with boiling treatment exhibiting the most significant increase. Boiled herbs demonstrated approximately 29% higher TPC and an impressive 348% increase in antioxidant activity compared to the other treatments. Oral administration of the boiled herb extracts to the HFD rats resulted in significant reductions in body weight, total cholesterol, triglycerides, and LDL cholesterol levels, while elevating the HDL cholesterol levels compared to the positive control rats. Additionally, the boiled herb extracts exhibited antioxidant, hepatoprotective, and nephroprotective effects. Notably, PE displayed more significant anti-obesity properties compared to ME, potentially due to higher TPC and antioxidant activity observed in PE compared to ME. In conclusion, this study highlights the potential positive effects of boiled parsley against obesity and recommends boiling treatment as the preferred method when heat treatment is required for herbs.

## 1. Introduction

Over the past few decades, obesity has emerged as a significant global health concern that poses risks to the overall well-being of populations worldwide [1]. Obesity can contribute to the development of various associated health conditions, including but not limited to type 2 diabetes, high blood pressure, and cardiovascular disorders [2]. The global prevalence of obesity has seen a significant rise, nearly tripling since the 1970s to 2016. During that year, more than 1.9 million adults aged 18 years and older were classified as overweight, with more than 650 million individuals experiencing severe obesity [3]. Obesity refers to the abnormal or excessive accumulation of fat in the body, which is associated with various health risks. The primary factor contributing to this excessive fat buildup is an imbalance between energy intake and energy expenditure [4]. A daily diet that is high in energy/calorie content, lacks sufficient portions of vegetables and fruits, and includes an abundance of sugary beverages and unhealthy food choices can greatly contribute to the excessive accumulation of body fat [5]. On the contrary, the daily consumption of vegetables has been linked in several studies to reduce the risk of obesity [6]. Vegetables are an excellent source of fiber, vitamins and minerals, and other bioactive components that are known to promote health-beneficial effects [7]. Green leafy vegetables, among all other types, are considered highly nutritionally rich containing large amounts of major micronutrients such as β-carotene, ascorbic acid, folic acid, riboflavin, calcium, and iron. In addition, they are considered extremely low in energy/calories, presenting an excellent food source for weight loss [8].

As an example of nutritive leafy vegetables, parsley (*Petroselinum crispum*) and mallow (*Corchorus olitorius*) are common types and widely consumed in several countries. Parsley leaves have been shown in several studies to promote a wide range of pharmacological effects, including hepatoprotective, hypotensive, antioxidant, and antidiabetic effects. These beneficial effects are derived mainly from the rich content of bioactive components present in parsley, including vitamins, minerals, essential oils, and polyphenols. Various types and forms of nutraceuticals were isolated from parsley, such as chlorophylls, carotenoids, phenolics, flavonoids, and non-flavonoid components [9,10]. The leaves of mallow are demonstrated as a rich source of several bioactive components such as vitamins, minerals, dietary fiber, fatty acids, and phenolics. Mallow leaves were reported to have a high content of omega-3 fatty acids, reaching almost 50% of the total leaves. Minerals such as iron, calcium, and zinc are also present in mallow leaves at relatively high concentrations compared to other types of vegetables [11].

Commonly, green leafy vegetables are widely consumed fresh, though evidence from many studies has shown that the nutrients’ bioavailability can be enhanced after cooking [7,12]. Cooking methods may significantly alter or impact the nutritional components of green leafy vegetables [12]. Cooking is a simple process that involves heating food to provide more edible and easily digested food [13]. The most common cooking methods for vegetables are steaming, boiling, frying, microwaving, and pressure cooking [12]. In light of this background, this study aims to investigate the effects of three cooking methods (blanching, boiling, and microwaving) on some of the bioactive nutrients of mallow and parsley. Depending on the findings of our study, the most effective cooking method to enhance the nutrients’ bioavailability will be further used in an experimental animal study to investigate their possible anti-obesity effects.

## 2. Materials and Methods

### 2.1. Ingredients

Fresh leaves of mallow (*Corchorus olitorius*, jute mallow cultivar) and parsley (*Petroselinum crispum* var. neapolitanum cultivar) were obtained from the local farmer’s market of Buraydah, Saudi Arabia. According to the botanical description of both plants [14,15], parsley was distinguished by its deeply divided, feathery leaves that are bright green in color. The leaves are arranged in a rosette pattern close to the ground. While mallow have round or lobed leaves that are often broader and have a softer texture compared to parsley.

### 2.2. Preparation of Samples

The fresh leaves of both herbs were thoroughly washed with distilled water (dH_2_O), finely chopped, and heat-treated separately using three methods, boiling, blanching, and microwaving. The boiling method was performed according to Abdalla and Yousef [16]. Briefly, the chopped leaves were mixed with dH_2_O in a ratio of 1:1 (*w*/*v*), covered in a pot, and heated to 100 °C for 10 min with regular stirring. For the blanching method, the chopped leaves were similarly mixed with dH_2_O in a ratio of 1:1 (*w*/*v*), covered in a pot, and heated to 90 ± 2 °C for 2 min with regular stirring according to Traoré et al. [17]. Lastly, for the microwaving method, 100 g of chopped leaves was mixed with 6 mL of dH_2_O and heated in a regular microwave oven for 1 min according to Turkmen et al. [18]. After cooling down, the heat-treated leaves were blended in a high-speed blender for 2 min to finely mince them, then freeze–dried and stored at −18 °C for further analysis.

### 2.3. Determination of Proximate Composition

The standard methods of AOAC were followed for the determination of the chemical composition, including moisture, ash, total solids, and pH values [19]. Dietary fiber was determined using the ANKOM fiber analyzer according to the AOAC (AOAC Method 991.43) [20]. Mineral content, including iron, zinc, magnesium, and calcium, was determined using atomic absorption spectroscopy according to the AOAC protocols [19].

### 2.4. Determination of Total Phenolic Content and Antioxidant Capacity

The samples were first extracted using 70% methanol (methanol:dH_2_O, 7:3, *v*/*v*) according to the extraction method of Abdalla and Yousef [16]. In brief, 1 g of freeze–dried sample was mixed with 100 mL of the 70% methanol, stirred regularly using a magnetic stirrer for 60 min, incubated at room temperature for a further 24 h, then centrifuged at 10,000× *g* for 20 min and the supernatant was filtered and stored at 4 °C. The total phenolic content (TPC) was determined following the Folin–Ciocalteu method according to Nsimba et al. [21]. Briefly, 20 μL of 70% methanolic extract was mixed with 100 μL of Folin–Ciocalteu’s reagent, incubated for 5 min at room temperature, and then 100 μL of 7.5% sodium carbonate (Na_2_CO_3_:dH_2_O, 7.5:92.5, *w*/*v*) was added. After 60 min of incubation in the dark at room temperature, the absorbance was measured at 765 nm using a microplate reader (Jenway UV-Vis spectrophotometer). For a standard curve, gallic acid solution was used for preparing a series of dilutions and results are expressed as mg gallic acid equivalents (GAE) per g of dry weight (mg GAE/g DW).

The antioxidant capacity was determined using the DPPH method according to Nsimba et al. [21]; briefly, 20 μL of 70% methanolic extract was mixed with 200 μL of DPPH solution and incubated in the dark at room temperature for 60 min, the absorbance was then measured at 517 nm using a microplate reader (Jenway UV-Vis spectrophotometer). For the control sample, the DPPH solution was mixed with methanol and the samples’ radical scavenging activity against DPPH was calculated following the formula:$$Antioxidant\;Capacity\;\left(\%\right)=\frac{Absorbance\;of\;Control-Absorbance\;of\;Sample}{Absorbance\;of\;Control}\times 100$$

### 2.5. Determination of Vitamin C

The content of vitamin C in fresh and treated samples was determined through direct titration with iodine according to a method described by Helmenstine [22]. Briefly, 2.5 g of freeze–dried sample was dissolved in 100 mL of dH_2_O, followed by centrifugation at 3000× *g* for 5 min. The resulting supernatant was then filtered and used for titration with a standard iodine solution, with the aid of a 1% starch indicator solution. The endpoint of the titration was determined by the appearance of a dark blue color. The standard iodine solution was prepared by dissolving potassium iodide and potassium iodate in dH_2_O, followed by the addition of 3 M sulfuric acid. The solution was then diluted to volume in a 500 mL volumetric flask. The standard ascorbic acid solution was prepared by dissolving an appropriate amount of ascorbic acid powder (0.250 g) in dH_2_O and then diluting it to volume in a 250 mL volumetric flask. The obtained results are expressed as milligrams per 100 g based on dry weight of the sample (mg/100 g DW).

### 2.6. Animals and Experimental Design

Twenty-four male albino Wistar rats weighing 120–140 g were obtained from the Animal House of Pharmacy College, King Saud University (Riyadh, Saudi Arabia) and housed at the Department of Food Science and Human Nutrition, College of Agriculture and Veterinary Medicine, Qassim University, Saudi Arabia, in polypropylene cages under standard laboratory conditions (22 ± 3 °C, 40–60% humidity, 12 h light/dark cycle) supplied with standard diet and water *ad libitum*. The standard diet (laboratory animals feed pellets) consists of 20% crude protein, 4% crude fat, 3.5% crude fiber, 6% ash, 0.5% salt, 1% calcium, 0.6% phosphorus, 20 IU/g vitamin A, 2.2 IU/g vitamin D, 70 IU/kg vitamin E, and 2850 ME Kcal/kg energy, with added trace minerals (cobalt, copper, iodine, iron, manganese, selenium, and zinc). The experiment was performed with the approval of the Committee of Research Ethics (Institutional Review Board, IRB) of Qassim University, Saudi Arabia (Approval No. 21-04-10).

After one week of adaptation, the rats were randomly divided into four groups, each consisting of six rats. The first group served as the negative control and was fed a standard diet. The remaining three groups were fed a high-fat diet (HFD) for six weeks to induce obesity. The HFD was a modified version of the standard diet, containing 20% protein, 32.7% carbohydrates, and 19.6% fats. After a six-week feeding period, the three groups on the HFD were reclassified as follows: one group served as the positive control group, while the other two groups were designated as treatment groups. All three groups maintained the HFD throughout the entire duration of the experiment. The treatment groups received oral gavage administration of freeze–dried boiled mallow and parsley aqueous extracts (ME and PE, respectively) at a dose of 200 mg/kg of body weight (BW) daily for eight weeks. The aqueous extracts were prepared by dissolving 2 g of freeze–dried sample in 100 mL of dH_2_O using a method described by El-Sahar [23]. At the end of eight week of oral administration, rats were fasted for 12 h, anesthetized with diethyl ether, and blood samples were collected through a cardiac puncture. Immediately after collection, the blood tubes were centrifuged at 3000× *g* at 10 °C for 15 min. The resulting serum samples were stored at −20 °C for subsequent biochemical analysis. Additionally, liver, kidney, and fat tissues were collected at the end of the experiment for histological examination.

### 2.7. Biochemical Analysis

#### 2.7.1. Determination of Fasting Blood Glucose and Serum Lipids

The collected serum samples were used to determine fasting blood glucose (FBG) according to the GOD-PAP method [24] by using an enzymatic colorimetric assay kit (Human Diagnostic, Wiesbaden, Germany). According to the manufacturer’s instructions, the serum samples were mixed with a reagent containing the enzyme glucose oxidase (GOD) to react with the samples’ glucose content leading to the production of hydrogen peroxide. The produced hydrogen peroxide further reacts under catalysis of peroxidase with phenol and 4-aminophenazone (PAP) leading to the formation of a red-violet colored compound and the intensity of the developed color was measured at 500 nm using a spectrophotometer (Jenway UV-Vis spectrophotometer). Serum lipids, including triglycerides, total cholesterol, and high-density lipoprotein cholesterol (HDL) were determined by using Human Diagnostic enzymatic colorimetric assays kits following the manufacturer’s protocols. Triglycerides and total cholesterol were measured according to the GPO-PAP and CHOD-PAP methods [24]. The absorbance of the generated colors was measured at 500 nm using a spectrophotometer. The levels of HDL were determined following a direct homogeneous assay [25], and the absorbance of the generated color was measured at 593 nm using a spectrophotometer. Further, the levels of low-density lipoprotein cholesterols (LDL) and very low-density lipoprotein cholesterol (VLDL) were calculated according to formulas of Friedewald et al. [26]. The levels of FBG and serum lipids are expressed as milligrams per deciliter (mg/dL).

#### 2.7.2. Determination of Liver Lipids

The concentrations of liver lipids, specifically total cholesterol and triglycerides, were determined using liver lipid extract, following the methods previously applied for serum samples [24]. The extraction and purification of the livers’ total lipids were carried out according to a method described by Folch et al. [27]. Briefly, the liver tissues were homogenized with a chloroform–methanol mixture in a ratio of 2:1. Subsequently, a washing step with dH_2_O was performed to remove non-lipid contaminants. This washing procedure resulted in the separation of two phases, with the lower phase containing the total pure lipid extract. The obtained results are expressed as milligrams per gram of wet weight (mg/g WW).

#### 2.7.3. Determination of the Livers’ Reduced Glutathione

The concentrations of the livers’ reduced glutathione (GSH) as a key oxidative stress marker was measured according to a method described by Moron et al. [28]. In brief, the diced liver tissues were initially homogenized using a Potter-Elvehjem homogenizer at a speed of 440 rpm. After homogenization, the liver tissues were diluted in water and then sonicated for 1 min. Trichloroacetic acid was added to the sonicated mixture, followed by centrifugation for 7 min. The supernatant was then used to react with 5,5′-dithiobis-(2-nitrobenzoic acid) (DTNB) and the absorbance of the generated yellowish colored compound was measured at 405 nm using a spectrophotometer. For a reference test, trichloroacetic acid at a concentration of 5% was used instead of the sample and the obtained results are expressed as micromoles per gram of wet weight (µmol/g WW).

#### 2.7.4. Determination of Kidney Functions

The concentrations of serum creatinine and urea, which are key markers for kidney function, were determined using Human Diagnostic assay kits following the manufacturer’s protocols. The levels of creatinine were measured using the Jaffé reaction method, which employs a photometric colorimetric test [29]. The absorbance of the resulting color was measured at 510 nm using a spectrophotometer. Urea levels on the other hand, were determined using the fully enzymatic method known as the CLDH method [30]. The absorbance of the generated color was measured at 546 nm using a spectrophotometer. The obtained results are expressed as milligrams per deciliter (mg/dL).

### 2.8. Histological Examination

The liver and kidney tissues obtained from three randomly selected rats in each group were preserved by immersing them in a 10% formalin solution (formalin:dH_2_O, 1:9, *v*/*v*) for a duration of 24 h. The tissues were then washed with tap water, dehydrated using a series of alcohol dilutions, and finally embedded in paraffin. The paraffin-embedded tissues were then subjected to baking at 56 °C for 24 h. To facilitate microscopic examination, four-micron sections were obtained from each tissue specimen. These sections were stained using hematoxylin and eosin (H&E) and observed under an electric light microscope [31]. The structural alterations were illustrated and diagnosed by an expert under a blind evaluation.

### 2.9. Statistical Analyses

The statistical analyses were performed using COSTAT software (version 6.4). To assess the differences between the means of the groups, Duncan’s test was applied. The results are expressed as mean ± standard deviation (SD), and a *p*-value below 0.05 was considered statistically significant.

## 3. Results

### 3.1. Proximate Composition

The chemical composition was evaluated in mallow and parsley leaves, fresh and heat-treated, as shown in Table 1. The moisture contents in the fresh parsley and mallow leaves were almost similar. After subjecting the samples to heat treatments, the moisture content exhibited a significant increase of 3.7–4.8% in parsley leaves and 4.6–7% in mallow leaves. The microwaved samples showed the highest increase in moisture content, followed by the blanched and boiled samples. Consequently, the treated parsley exhibited a decrease in total solids of 27%, 25%, and 97.6% for blanched, boiled, and microwaved samples, respectively. Similarly, the treated mallow showed a decrease in total solids of 46%, 30.65%, and 99.92% for blanched, boiled, and microwaved mallow, respectively. Fresh samples of parsley also recorded the highest concentration of ash content compared to treated samples, while the ash content of mallow recorded nonsignificant changes between the fresh and the treated samples. Dietary fiber content also showed nonsignificant changes after the three cooking methods as shown in Table 1.

### 3.2. Mineral Content

The mineral content of parsley and mallow, including iron, zinc, magnesium, and calcium, was determined in the fresh and heat-treated samples as shown in Table 2. Parsleys’ iron content showed nonsignificant changes after the boiling method compared to the fresh sample. However, the other treatments, blanching and microwaving, resulted in a nonsignificant decrease in the iron content of parsley samples compared to the fresh samples. For the mallows’ iron content, boiling and microwaving had no effects in changing the content compared to the fresh samples, whereas the iron content after the blanching treatment recorded an increase compared to the fresh samples. However, these results were also statistically insignificant. The zinc content of the blanched and microwaved parsley samples showed a significant decrease of 34.76% and 43%, respectively, compared to the fresh samples. Whereas the boiling method had nonsignificant effects on the parsleys’ zinc content. For the mallows’ zinc content, all three heat treatments led to significant decreases compared to the fresh samples. The magnesium content in the parsley samples also showed a significant decrease after microwaving compared to the fresh samples, whereas for the other heat treatments, blanching and boiling, no changes were recorded compared to the fresh samples. The mallows’ magnesium content also showed consistent results; the microwaving treatment was the only treatment in showing a significant decrease of nearly 3% in the magnesium content compared to the fresh samples. Lastly, the calcium content of parsley samples showed significant decreases after the three heat treatments compared to the fresh samples of 30%, 11%, and 20% in the blanched, boiled, and microwaved samples, respectively. These results notably indicate that boiled parsley was the sample showing the least reduction compared to the other treatments. On the other hand, the mallow’s calcium showed an increase in its content compared to the other treatments, even to the fresh samples. An increase of 14% was recorded in calcium content of the blanched mallow leaves. While the other treatments showed a reduction of 6–7%.

### 3.3. Vitamin C, Total Phenolics, and Antioxidant Capacity

The content of vitamin C in the fresh and treated samples was determined as shown in Table 3. All three heat treatments resulted in a reduction in the vitamin C content of the parsley samples compared to the fresh samples, with reductions of 2.56%, 7.69%, and 48.72% for blanching, boiling, and microwaving, respectively. It is evident that the microwaving treatment had the highest impact on reducing the vitamin C content of parsley compared to the other treatments, nearly reaching a 50% decrease compared to the fresh samples. Microwaved mallow samples also showed a significant reduction in the vitamin C content, however, not as much as that noted in parsley samples. All three treatments significantly decreased the content by 6–20% compared to the fresh samples.

The TPC content of both parsley and mallow alcoholic extracts, as shown in Table 3, showed significant increases of 23–28% and 225–347%, respectively. The boiling treatment was the highest in showing such significant increases for both parsley and mallow leaves. The antioxidant capacity also recorded significant increases after the three treatments compared to the fresh samples as shown in Table 3. Boiling and microwaving treatments were the most effective in increasing the capacity for both parsley and mallow leaves by 82–89% and 607–616%, respectively.

### 3.4. Experimental Study

#### 3.4.1. Body Weight and Food Intake

The initial body weight of the HFD-fed rats was significantly higher than that of the normal rats due to the daily intake of the HFD for six weeks (Table 4). The administration of ME and PE at a dose of 200 mg/kg BW for eight weeks resulted in lower body weight compared to the positive control group. The final body weight of the positive control group was the highest, followed by the HFD + ME and HFD + PE. The latter group, HFD + PE, had a nonsignificant difference in the final body weight compared to the negative control group. When comparing the percentage of body weight gain to the positive control group, both the negative control and HFD + ME showed a significant reduction of 40%, while the HFD + PE exhibited an even greater reduction of 80.57%. In other words, the PE treatment demonstrated the highest effectiveness in reducing the body weight percentage compared to all other groups. Moreover, the food intake of the HFD + ME and HFD + PE groups was lower by 14% and 17%, respectively, compared to the positive control group. However, the observed reduction was not statistically different from that of the positive control group.

#### 3.4.2. Relative Size of Liver, Kidneys, and Adipose Tissue

As shown in Table 5, the liver weight of the positive control group was 15% higher compared to the negative control group. On the contrary, both treated groups, HFD + ME and HFD + PE, had similar weights compared to the negative control. The kidney size of the positive control group was also the highest compared to all groups, whereas the two treated groups were similar to the negative control (Table 5). However, these results were statistically nonsignificant. Further, the positive control group also had more adipose tissue compared to the other groups as shown in Table 5. The HFD + PE group was the only group that showed a significant reduction in adipose tissue compared to the positive control group, with a decrease of nearly 60%.

#### 3.4.3. Blood Glucose and Serum Lipids

The fasting blood glucose levels measured at the end of the experiment were significantly higher in the positive control group compared to the negative control group (Table 6). In contrast, the HFD-fed rats administered with ME and PE had significantly lower glucose levels compared to the positive control group, with reductions of 19% and 28%, respectively. The two extracts also had some significant differences in alleviating the blood glucose levels; the HFD + PE had significantly lower glucose levels compared to the HFD + ME as shown in Table 6.

The total cholesterol levels were also significantly higher in the positive control group compared to the negative control (Table 6). The administration of both ME and PE resulted in a significant reduction in the total cholesterol levels compared to the positive control group, with reductions of 24.64% and 39.22%, respectively. The HFD + PE had significantly lower total cholesterol levels compared to the HFD + ME, with no significant difference compared to the negative control group. The triglyceride levels were also significantly elevated in the positive control group compared to the other groups. PE extract administration resulted in a significant reduction in the triglycerides compared to the positive control group as well as to the HFD + ME group (Table 6). The level of HDL in the positive control group was significantly lower compared to the other groups. Following the administration of both extracts, HFD + PE recorded an increase to the normal levels as shown in Table 6, with PE consistently showing slightly more effectiveness compared to the ME (26.15 ± 2.9 mg/dL vs. 24.12 ± 0.9 mg/dL in the HFD + PE vs. HFD + ME, respectively). However, these results showed nonsignificant differences. The two extracts showed significant differences in terms of reducing the levels of LDL and VLDL cholesterols; although the administration of both extracts had consistent results in reversing the LDL and VLDL to normal levels, the PE was significantly more effective in lowering the LDL and VLDL levels compared to the ME as shown in Table 6.

#### 3.4.4. Liver Lipids

The liver lipids, including total cholesterol and triglycerides, were significantly higher in the positive control group compared to the negative control group (Table 7). The administration of both extracts, ME and PE, resulted in a significant reduction in liver cholesterol and triglycerides of 35–72% and 50–67%, respectively. Both extracts had nonsignificant differences compared to the negative control group. Although consistent with previous results, the PE treatment was more effective than ME in reducing liver lipids. Differences of 56.79% and 29.80% were observed in the liver cholesterol and triglyceride concentrations, respectively, following treatment with PE compared to ME. Furthermore, the concentration of total liver fats was higher in the positive control group compared to the other groups. The administration of ME and PE significantly reduced the liver fat concentration by 59.22% and 60.82%, respectively.

#### 3.4.5. Oxidative Stress Markers

The reduced glutathione (GSH) was determined in the rats’ liver tissues as shown in Table 7. The positive control group showed a significant decrease in liver glutathione when compared to the negative control (11.87 to 6.43 µmol/g WW). The administration of both extracts, ME and PE, significantly increased the glutathione concentrations to the normal levels, indicating their efficiency in reversing oxidative stress to the normal state.

#### 3.4.6. Kidney Functions

The concentrations of creatinine and urea were measured in the experimental rats as presented in Table 8; the positive control group had significantly higher concentrations compared to the negative group. In contrast, the HFD + ME and HFD + PE showed lower concentrations of creatinine and urea compared to the positive control group by 29–36% and 10%, respectively. The concentrations also had nonsignificant differences compared to the negative control group indicating the efficiency of both extracts in promoting nephroprotective effects.

#### 3.4.7. Histological Examination

The hepatic tissues were histologically examined under a microscope (Leica DMD108) as shown in Figure 1. The negative control group had normal hepatic structure compared to the positive control (Figure 1A,B). The positive control group had extreme dysfunctions in the hepatic structure presented by steatosis of hepatocytes with severe congestion in the central vein. In contrast, Figure 1C shows fewer hepatocytes with steatosis and mild congestion, and Figure 1D shows a similar structure presented by mild congestion. These results indicate that the administration of both ME and PE preserved the hepatic structure. Furthermore, the nephritic structure was also subjected to histological examination as shown in Figure 2. The negative control group had normal glomerular architecture and tubule morphology with no signs of any damage (Figure 2A). In contrast, the positive control group had severely abnormal glomerular architecture, destruction and necrosis of renal tubules, detachment of the basement membrane, and severe congestion (Figure 2B). The administration of both ME and PE resulted in normal nephritic structures as shown in Figure 2C,D, indicating that this intervention preserved the nephritic cells from the HFD complications.

## 4. Discussion

In recent decades, green leafy vegetables have been demonstrated as highly beneficial for the prevention of obesity due to their high fiber content as well as due to the presence of various bioactive components. Heat exposure or cooking methods applied in the food industry may positively or negatively affect the presence of such bioactive substances. In the current work, three heat treatments (blanching, boiling, and microwaving) were applied to parsley and mallow leaves and the results showed significant changes in the nutrients’ bioavailability. The three heat treatments had interestingly resulted in more TPC and antioxidant activity compared to fresh samples; the boiling treatment was the most effective in showing such positive results.

To investigate the anti-obesity effects, alcoholic extracts of boiled parsley and mallow leaves were chosen for further biological study in obese rats. Firstly, the three heat treatments resulted in multiple changes in the composition of parsley and mallow leaves. The moisture content increased to 90–92% compared to nearly 87% recorded in the fresh leaves. Similar moisture content in the fresh parsley and mallow leaves was reported in multiple studies [16,32,33,34]. The increase in moisture content is mainly due to the exposure to high temperatures that leads to destruction in the cell walls resulting in more water absorption [17]. Consequently, the total solids for the treated samples showed a slight decrease from 13% in the fresh samples to 9–7%. On the other hand, the fiber content showed nonsignificant changes after the three treatments. Both fresh parsley and mallow leaves were found to have excellent amounts of fiber initially. Fresh parsley had a fiber content of 24.27 ± 0.00 g/100 g DW, similar to the reported fiber content of 30.41 g/100 g in fresh parsley grown in Egypt [35]. Fresh mallow leaves, on the other hand, exhibited a higher fiber content of 32.77 ± 0.00 g/100 g DW. This contrasts with other studies that reported lower fiber content ranging from 5 to 12 g/100 g in fresh mallow leaves [16,34]. The treated parsley and mallow samples had similar fiber contents compared to the fresh samples in amounts ranging from 23.50 to 23.77 and 32.60 to 32.89 g/100 g, respectively. Although treatments like boiling and microwaving can lead to a loss in fiber content, the degree of fiber loss can vary depending on several factors, including time and temperature. Shorter cooking times and lower temperatures can minimize the loss in fiber in vegetables [36]. The applied treatments in the current work did not exceed 10 min for boiling and 1–2 min for microwaving and blanching, and hence, limited heat exposure might be linked to the maintenance of fiber content. On the other hand, it is worth noting that some types of fiber are more heat-resistant than others. Insoluble fiber, such as cellulose and hemicellulose, tends to be more resistant to heat compared to soluble fiber [37,38]. Therefore, the specific composition of fiber in parsley and mallow leaves might also influence its response to different heat treatments. Adequate fiber intake is important for preventing diseases like obesity and type 2 diabetes [39]. The nonabsorbable fiber fractions like cellulose and hemicellulose can promote healthy digestion by increasing gut volume and improving motility, which helps remove toxins more efficiently [8,40,41].

Furthermore, green leafy vegetables are widely recognized as a valuable source of essential minerals, including iron, calcium, and zinc. In the current study, both fresh parsley and mallow leaves demonstrated significant levels of these minerals, with some aligning with the values reported by the USDA [42]. Our findings were consistent with the results reported in other studies, further supporting the observed mineral content in fresh leaves [33,34,43]. The heat treatments applied in the current work resulted in changes in mineral content, with varying effects on the iron and zinc content of parsley and mallow leaves. Blanching and microwaving led to a decrease in iron content in parsley leaves, while boiling did not significantly affect the iron content in either type of leaf. Zinc content remained unchanged in parsley leaves after boiling, but boiled mallow leaves showed a decrease in zinc content. Minerals can leach out into the cooking liquid leading to their loss, however, the leaching can be influenced by factors such as time, temperature, and water volume [44,45]. Although both liquid treatments used in our study employed a similar volume and almost similar temperature, and despite the longer duration of boiling treatment that showed higher iron content, the varying effects observed in the iron content might be attributed to other factors such as oxidation. Minerals such as iron are known to be susceptible to oxidation and degradation when exposed to air during the cooking process [46]. Thus, it is reasonable to consider that inadvertent delays in promptly covering the treated plants could have potentially contributed to the observed variations in results. Further, the calcium content of parsley and mallow leaves also showed a significant reduction of 11–30% and 6–7%, respectively, after the three treatments. These varying results might relate to the differences in texture or structure of the two types of leaves. Alterations in texture or structure of leafy vegetables during heat processing can impact the availability and leaching of minerals [47,48]. Consequently, changes in mineral content after exposure to high temperatures were consistently observed in multiple studies [13,16]. On the other hand, the observed changes in minerals are consistent with the changes in ash content, as lower concentrations of ash were observed after the treatments compared to untreated samples. The ash content of a food reflects its mineral content, as the residual ash remaining after incineration primarily consists of the inorganic minerals originally found in the food [49].

On the other hand, vitamin content in green leafy vegetables can similarly show some changes after applying different heat treatments. Vitamin C, in particular, is highly sensitive to heat [50]. Fresh samples of parsley were found to contain 100 mg/100 g DW of vitamin C, which is partially consistent with the findings of Bediar [51], who reported 133 mg/100 g of vitamin C in fresh parsley leaves. Fresh mallow leaves showed a lower amount of vitamin C (77 mg/100 g DW) aligning with the results reported in other studies [33,52]. The heat treatments resulted in a significant reduction in the vitamin C content of both leaves. Considering the sensitivity of vitamin C to high temperatures [50], these results were not surprising. Similar outcomes have been reported in other studies where green leafy vegetables showed consistent reductions in vitamin C content after being exposed to high cooking temperatures [13].

Green leafy vegetables also contain significant amounts of bioactive components such as phenolic compounds which are well known for promoting several health-beneficial effects, including anti-inflammatory, anticancer, antimicrobial, and antioxidant effects [53]. The consumption of a daily diet rich in phenolic compounds has been demonstrated to promote anti-obesity and antidiabetic effects [54]. In the current work, the TPC exhibited a significant increase following the three heat treatments, with parsley showing a rise of 23–29%, while mallow exhibited a substantial increase of 225–347%. It is worth noting that mallow leaves initially had lower TPC levels in the fresh samples compared to parsley. This disparity in initial TPC levels could explain the more pronounced increase observed in mallow after the heat treatments when compared to parsley. It has been reported that fresh mallow leaves contain TPC ranging from 5.56 to 42.3 mg GAE/g [55]. In our observation, we found the TPC content in fresh mallow fell within this range (36.26 ± 3.88 mg GAE/g). The fresh parsley, on the other hand, showed consistent results with the findings of El-Sayed et al. [56], who reported a TPC content of 121.95 mg GAE/g in fresh Egyptian parsley leaves. In the current study, the TPC increased after the heat treatments to 184.87 mg GAE/g and 162.38 mg GAE/g in the treated parsley and mallow, respectively. Although some phenolic compounds can be degraded or lost during cooking, others can be released or transformed into more bioavailable forms. Heat treatments can facilitate the extraction of phenolic compounds from plant tissues due to the breakdown of plant cell walls and membranes, which can result in the release of phenolics that are bound or sequestered within the cells [57,58]. It was indicated by Ferracane et al. [59] that treatments such as boiling can result in thermal disruption of the non-covalent bonds linking phenolic components to the cells’ chloroplast proteins, consequently leading to more extractability of phenolics. In addition, processes such as boiling, blanching, or steaming can result in water loss from foods, as evidenced by the observed reduction in moisture content in the current study. The decrease in moisture content can contribute to higher concentrations of phenolic compounds. As the water content decreases, the phenolic compounds become more concentrated, thereby leading to an increased TPC per unit weight or volume of the cooked leaves [57,60].

In line with the significant increase in TPC, the antioxidant capacity measured in the treated samples accordingly showed an increase compared to fresh samples. Fresh mallow leaves initially had an antioxidant capacity of 12% less than that reported in other studies that found 22% of antioxidants in fresh mallow leaves [61]. The applied heat treatments in the current work resulted in an increase in the antioxidant capacity to 40–88%, with the boiling method consistently resulting in the most significant increase compared to blanching and microwaving. Fresh parsley leaves, on the other hand, in the current work showed an antioxidant capacity of 25%, similar to that reported in some studies that found 30% of antioxidants in fresh parsley [62]. After the heat treatments, the parsley’s antioxidants increased to 29–47% and the boiling method consistently showed the highest effectiveness. This enhanced antioxidant activity can be attributed to the presence of the phenolic compounds, such as polyphenols, which exhibit potent antioxidant properties by effectively scavenging and neutralizing harmful free radicals within the physiological milieu [63,64]. Nonetheless, it should be taken into concern that varying temperature degrees may show different results; frying processes, for instance, were reported to promote a lower antioxidant capacity in green leafy vegetables compared to boiling or steaming. The latter processes were demonstrated to promote higher polyphenols and flavonoids compared to fresh samples [13,50,65]. Therefore, applying appropriate cooking methods can significantly enhance or alter nutrients’ bioavailability.

In light of these findings, the boiled parsley and mallow’s alcoholic extracts, PE and ME, were further administrated to HFD-fed rats to investigate their possible anti-obesity effects. The extracts were orally administrated at 200 mg/kg BW in combination with HFD feeding, and the results showed significant weight loss of 7–14% compared to the positive control group rats. A significant reduction in fat accumulation was also observed based on decreased fat tissue sizes by 38–57% compared to the positive control group. However, the most effective results in preventing weight gain were noted in the HFD + PE group. Despite the fact that the boiled mallow leaves had a higher fiber content compared to that of boiled parsley, the TPC content of the latter was higher. Phenolic compounds were reported to promote anti-obesity effects by playing roles in the regulation of adipocyte metabolism leading to the prevention of fat accumulation [2,23], which could explain the more favorable results obtained after the administration of PE compared to ME.

Furthermore, excessive weight gain is commonly linked to abnormalities in serum glucose and lipid levels [66]. In the current work, feeding with HFD resulted in significant increases in the levels of FBG, total cholesterol, triglycerides, and LDL with a significant reduction in HDL levels compared to the negative control group. After the administration of both PE and ME, the levels of FBG showed a significant alleviation compared to the positive control group. In this respect, PE showed a higher effectiveness than ME; the higher TPC in the PE may also explain its effectiveness in alleviating the FBG. Phenolic compounds such as myricetin, one of the most important flavonoid components in parsley, has been demonstrated to play roles in alleviating blood glucose by enhancing insulin sensitivity. It was shown in a study conducted in diabetic rats that the administration of parsley leaves for four weeks led to a significant reduction in serum glucose [67]. Serum lipids also showed significant enhancements after the administration of PE in the current work; total cholesterol, triglycerides, LDL, and HDL levels changed significantly and reversed to normal levels. Parsley contains significant amounts of bioactive components such as tocopherols, β-pinene, terpinolene, and apiin that were demonstrated in several studies to promote cholesterol-lowering properties [23,68,69,70,71]. Mallow leaves can be also considered beneficial in promoting glucose and cholesterol-lowering properties based on the positive results obtained in the current work. Other studies also showed beneficial effects after the administration of mallow leaves; in streptozotocin-fed diabetic rats, the administration of mallow leave methanolic extract at doses of 100 and 200 mg/kg BW resulted in improvements in glucose levels [72]. In another study conducted in alloxan-fed diabetic rats, mallow leave powder was mixed with the rats’ chow for 14 days and results showed significant improvements in the levels of serum glucose, cholesterol, triglycerides, and LDL cholesterol. It was indicated that mallow’s beneficial effects are attributed to the presence of flavonoids, alkaloids, and glycosides [73]. Others reported that mallow leaves are also rich in saponins that are known to promote cholesterol-lowering properties by inhibiting its intestinal absorption via the formation of a non-absorbable complex with cholesterol [2,74,75]. In addition, green leafy vegetables in general were demonstrated to promote cholesterol-lowering properties mediated by their excellent fiber content [8]. Mallow leaves had an excellent amount of fiber after the boiling method; fiber can boost the gut microbiota leading to the production of multiple bioactive substances as byproducts such as short-chain fatty acids that are known as cholesterol absorption inhibitors [8,40,41].

Changes in liver lipids were screened in the current work; the liver plays a significant role in the regulation of serum cholesterol via its synthesis in hepatocytes [76]. Feeding with HFD resulted in significant abnormalities in liver lipids in the positive control group compared to the negative control. The administration of both extracts, PE and ME, resulted in a significant reduction in liver cholesterol and triglycerides compared to the positive control group of 35–70 and 52–67%, respectively. Consistent with earlier results, PE had the most effectiveness in promoting such improvements in liver lipids compared to ME. These positive effects can relate to the higher TPC and their related antioxidants recorded in the PE. Oxidative stress, the imbalance in the production of free radicals, is well demonstrated in playing key roles in the promotion of various abnormalities in the body’s biological functions [77]. The development of obesity complications has been linked directly or indirectly to oxidative stress [54]. Phenolic compounds can promote anti-obesity effects via their strong antioxidant properties. In the current work, the abnormalities observed in liver lipids of the positive control group were accompanied by a significant reduction in the level of lipids’ reduced glutathione, a major physiological antioxidant. Accordingly, the administration of both extracts resulted in reversing the decrease in the lipids’ glutathione activity of 77–88%, which could explain the enhancements in liver lipid levels [76]. Parsley and mallow leaves, in general, contain rich amounts of several types of antioxidant substances as mentioned earlier; antioxidants can significantly promote the liver’s lipid metabolism leading to the prevention of fat accumulation [66].

Kidney function was also assessed in the current work to investigate the possibility of both extracts to promote nephroprotective effects. Compared to the positive control group, the administration of both extracts had sufficiently reversed the imbalances in creatinine and urea concentration that occurred as a result of the HFD feeding. Parsley’s myricetin and apiol are demonstrated to play roles in reducing the body’s toxins by increasing the kidneys’ filtration [78,79]. Similar enhancements in the kidney’s creatinine and urea concentrations after the administration of a mixture of green plants, including mallow leaves, were reported by El-Sahar [23]. Such imbalances in liver lipids and kidney functions observed in the positive control group can reflect on the tissue of both organs. The administration of both extracts also showed some beneficial effects on the hepatic and nephritic structures as observed by performing a histological examination. Feeding with a HFD had notably resulted in massive steatosis of hepatocytes with severe congestion in the central vein, while the kidney structure had a severely abnormal glomerular architecture presented by destruction and necrosis of renal tubules, detachment of the basement membrane, and severe congestion. Such dysfunctions and abnormalities were not noted in the tissues of the HFD + PE and HFD + ME groups, indicating their beneficial effects in promoting hepatoprotective and nephroprotective effects.

## 5. Conclusions

The findings of this study showed that applying three heat treatments (blanching, boiling, and microwaving) can significantly promote changes in the nutritional and functional aspects of two green leafy vegetables, parsley and mallow. The significant changes after the heat treatments were presented by increases in the total phenolic content and their related antioxidant activity. The boiling method was the most effective in showing such positive results. The oral administration of boiled parsley and mallow extracts at a dose of 200 mg/kg BW to high-fat-diet-fed rats resulted in significant anti-obesity effects based on improvements in the levels of serum glucose and lipids, as well as improvements in the liver lipids, kidney functions, and the structure of liver and kidney tissues. These positive results were significantly attributed to the prevention of oxidative stress as noted in enhancements in the concentrations of the livers’ reduced glutathione. Notably, parsley leaves were more effective in promoting anti-obesity effects compared to mallow. Nonetheless, the regular consumption of parsley and mallow leaves is highly recommended for the prevention of obesity.

## Figures and Tables

**Figure 1 foods-12-04303-f001:**
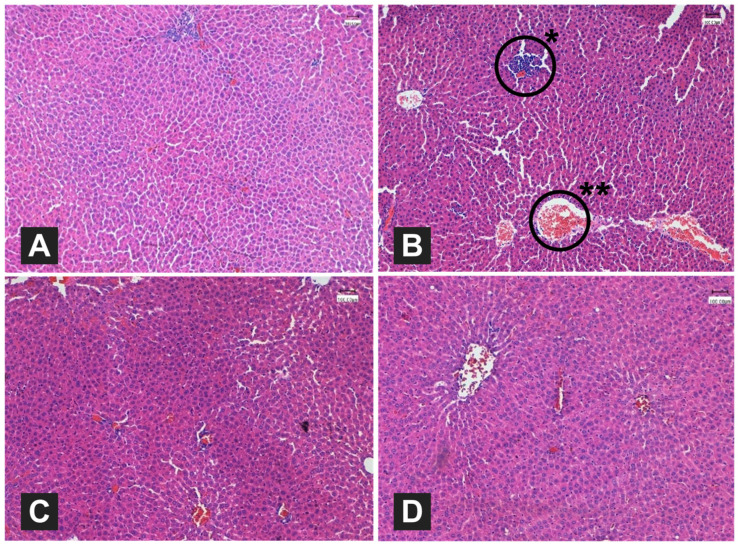
Histological examination of the hepatic structure of high-fat diet (HFD)-fed rats administrated orally with boiled mallow and parsley extracts (200 mg/kg BW) (H&E stain ×100). (**A**) Negative control group; (**B**) positive control group; (**C**,**D**) HFD-fed rats orally administrated with mallow and parsley extracts, respectively. *: leukocytes obstructed around blood vessels; **: extreme obstruction of leukocytes.

**Figure 2 foods-12-04303-f002:**
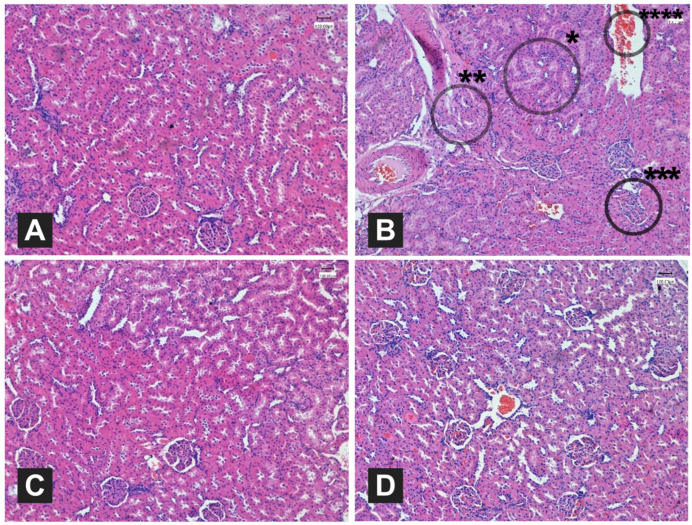
Histological examination of kidney tissues of high-fat diet (HFD)-fed rats administrated orally with boiled mallow and parsley extracts (200 mg/kg BW). (**A**) Negative control group; (**B**) positive control group; (**C**,**D**) HFD-fed rats orally administrated with mallow and parsley extracts, respectively. *: cloudy swelling; **: destruction and necrosis of renal tubules; ***: abnormal glomerular architecture; ****: leukocyte obstruction with severe congestion.

**Table 1 foods-12-04303-t001:** Proximate composition of parsley and mallow leaves, fresh and heat-treated.

Parameter	Parsley				Mallow			
	Fresh	Blanched	Boiled	Microwaved	Fresh	Blanched	Boiled	Microwaved
Moisture (%)	86.97 ± 0.49 ^b^	90.54 ± 0.67 ^a^	90.24 ± 0.02 ^a^	91.14 ± 0.31 ^a^	86.95 ± 0.06 ^c^	92.95 ± 0.06 ^a^	90.95 ± 0.02 ^b^	92.68 ± 0.01 ^a^
TDS (%)	13.03 ± 0.4 ^a^	9.46 ± 0.67 ^b^	9.76 ± 0.02 ^b^	8.86 ± 0.31 ^b^	13.05 ± 0.06 ^a^	7.04 ± 0.06 ^c^	9.05 ± 0.02 ^b^	7.32 ± 0.01 ^c^
Ash (%)	12.94 ± 0.9 ^a^	8.95 ± 1.86 ^c^	11.83 ± 0.41 ^b^	9.8 ± 0.52 ^bc^	16.15 ± 0.04 ^a^	16.38 ± 0.08 ^a^	15.48 ± 0.67 ^a^	15.91 ± 0.12 ^a^
Dietary Fiber (g/100 g) *	24.27 ± 0.00 ^a^	23.79 ± 0.00 ^b^	23.51 ± 0.00 ^d^	23.66 ± 0.00 ^c^	32.77 ± 0.00 ^c^	32.87 ± 0.00 ^b^	32.89 ± 0.00 ^a^	32.65 ± 0.00 ^d^

* Based on DW (dry weight). TDS: total dissolved solids. Results expressed as mean ± SD; different superscripted letters (a, b, c, d) in the same row indicate a significant difference between means at *p* < 0.05.

**Table 2 foods-12-04303-t002:** Mineral content (mg/100 g DW) of parsley and mallow leaves, fresh and heat-treated.

Element	Parsley				Mallow			
	Fresh	Blanched	Boiled	Microwaved	Fresh	Blanched	Boiled	Microwaved
Fe	13.92 ± 0.38 ^a^	8.28 ± 0.21 ^b^	12.18 ± 0.86 ^a^	7.36 ± 0.52 ^b^	8.61 ± 0.15 ^b^	11.75 ± 0.59 ^a^	7.15 ± 0.62 ^b^	8.11 ± 1.64 ^b^
Zn	1.64 ± 0.03 ^a^	1.07 ± 0.04 ^b^	1.57 ± 0.07 ^a^	0.93 ± 0.07 ^c^	0.84 ± 0.01^a^	0.53 ± 0.04 ^b^	0.36 ± 0.05 ^c^	0.39 ± 0.07 ^bc^
Mg	181.52 ± 1.36 ^a^	182.8 ± 0.47 ^a^	182.62 ± 0.29 ^a^	179.63 ± 1.43 ^b^	185.49 ± 0.65 ^a^	184.89 ± 0.28 ^a^	184.52 ± 2.89 ^a^	180.35 ± 0.69 ^b^
Ca	325.35 ± 1.20 ^a^	227.58 ± 2.18 ^d^	287.83 ± 1.39 ^b^	259.61 ± 3.41 ^c^	594.68 ± 4.45 ^b^	694.1 ± 4.92 ^a^	558.51 ± 2.32 ^c^	548.74 ± 0.14 ^d^

Fe: iron; Zn: zinc; Mg, magnesium; Ca: calcium. Results expressed as mean ± SD; different superscripted letters (a, b, c, d) in the same row indicate a significant difference between means at *p* < 0.05.

**Table 3 foods-12-04303-t003:** Vitamin C content, total phenolic content, and antioxidant capacity of parsley and mallow leaves, fresh and heat-treated (based on DW).

Parameter	Parsley				Mallow			
	Fresh	Blanched	Boiled	Microwaved	Fresh	Blanched	Boiled	Microwaved
Vitamin C (mg/100 g)	100 ± 0.00 ^a^	97.44 ± 0.00 ^b^	92.31 ± 0.00 ^c^	51.28 ± 0.00 ^d^	76.92 ± 0.00 ^a^	71.79 ± 0.00 ^b^	66.67 ± 0.00 ^c^	61.54 ± 0.00 ^d^
TPC (mg GAE/g)	143.35 ± 1.80 ^c^	178.3 ± 3.96 ^b^	184.87 ± 3.67 ^a^	176.73 ± 5.11 ^b^	36.26 ± 3.88 ^d^	117.85 ± 1.29 ^c^	162.38 ± 2.64 ^a^	135.64 ± 1.65 ^b^
AOC (%)	25.18 ± 1.88 ^b^	29.79 ± 1.03 ^b^	47.83 ± 2.52 ^a^	45.87 ± 0.29 ^a^	12.34 ± 2.77 ^c^	43.06 ± 3.68 ^b^	88.41 ± 0.45 ^a^	87.33 ± 0.49 ^a^

DW: dry weight; TPC: total phenolic content; AOC: antioxidant capacity. Results expressed as mean ± SD; different superscripted letters (a, b, c, d) in the same row indicate a significant difference between means at *p* < 0.05.

**Table 4 foods-12-04303-t004:** The efficiency of boiled parsley and mallow leave extracts (200 mg/kg BW, oral administration) on body weight and food intake in HFD-fed rats, *n* = 6.

Groups	Initial Weight (g)	Final Weight (g)	Weight Gain (g)	Weight Gain (%)	Food Intake (g/Day)
NC	269.15 ± 10.73 ^b^	303.8 ± 12.51 ^c^	34.1 ± 1.17 ^b^	12.68 ± 0.82 ^b^	19.43 ± 2.75 ^a^
HFD	308.5 ± 7.8 ^a^	373.8 ± 8.03 ^a^	65.3 ± 4.8 ^a^	21.2 ± 1.8 ^a^	17.87 ± 3.41 ^ab^
HFD + ME	307.7 ± 28.0 ^a^	345.4 ± 15.01 ^b^	37.7 ± 17.8 ^b^	12.7 ± 6.5 ^b^	15.39 ± 3.29 ^b^
HFD + PE	307.8 ± 26.49 ^a^	319.3 ± 15.8 ^c^	11.4 ± 5.8 ^c^	4.12 ± 2.4 ^c^	14.85 ± 2.96 ^b^

NC: negative control; HFD: high-fat diet; HFD + ME: HFD + mallow extract; HFD + PE: HFD + parsley extract. Results expressed as mean ± SD; different superscripted letters (a, b, c) in the same column indicate a significant difference between groups’ means at *p* < 0.05.

**Table 5 foods-12-04303-t005:** The efficiency of boiled parsley and mallow leave extracts (200 mg/kg BW, oral administration) on liver, kidneys, and adipose tissues’ relative size of HFD-fed rats, *n* = 6.

Groups	Liver (g)	Kidneys (g)	Adipose Tissue (g)
NC	7.88 ± 0.82 ^ab^	1.67 ± 0.10 ^a^	2.22 ± 1.23 ^ab^
HFD	9.07 ± 0.30 ^a^	2.07 ± 0.11 ^a^	4.92 ± 0.69 ^a^
HFD + ME	7.05 ± 0.70 ^b^	1.89 ± 0.15 ^a^	3.02 ± 1.52 ^ab^
HFD + PE	7.75 ± 0.34 ^ab^	1.76 ± 0.21 ^a^	2.07 ± 1.09 ^b^

NC: negative control; HFD: high-fat diet; HFD + ME: HFD + mallow extract; HFD + PE: HFD + parsley extract. Results expressed as mean ± SD; different superscripted letters (a, b) in the same column indicate a significant difference between groups’ means at *p* < 0.05.

**Table 6 foods-12-04303-t006:** The efficiency of boiled parsley and mallow leave extracts (200 mg/kg BW, oral administration) on serum glucose and lipids (mg/dL) in HFD-fed rats, *n* = 6.

Groups	FBG	TC	Triglycerides	HDL	LDL	VLDL
NC	78.4 ± 0.9 ^d^	60.45 ± 1.2 ^c^	47.18 ± 1.3 ^c^	28.49 ± 2.7 ^a^	22.53 ± 1.6 ^c^	9.44 ± 0.3 ^c^
HFD	116.73 ± 2.2 ^a^	99.01 ± 2.8 ^a^	89.18 ± 4.6 ^a^	19.23 ± 3.4 ^b^	61.95 ± 6.5 ^a^	17.84 ± 0.9 ^a^
HFD + ME	94.49 ± 3.7 ^b^	74.61 ± 3.8 ^b^	68.36 ± 2.2 ^b^	24.12 ± 0.9 ^ab^	36.82 ± 4.1 ^b^	13.67 ± 0.4 ^b^
HFD + PE	83.78 ± 0.0 ^c^	60.18 ± 4.7 ^c^	46.63 ± 2.9 ^c^	26.15 ± 2.9 ^a^	24.7 ± 4.4 ^c^	9.33 ± 0.6 ^c^

NC: negative control; HFD: high-fat diet; HFD + ME: HFD + mallow extract; HFD + PE: HFD + parsley extract; FBG: fasting blood glucose; TC: total cholesterol; HDL: high-density cholesterol; LDL: low-density cholesterol; VLDL: very low-density cholesterol. Results expressed as mean ± SD; different superscripted letters (a, b, c, d) in the same column indicate a significant difference between groups’ means at *p* < 0.05.

**Table 7 foods-12-04303-t007:** The efficiency of boiled parsley and mallow leave extracts (200 mg/kg BW, oral administration) on liver lipids and glutathione in HFD-fed rats, *n* = 6.

Groups	Cholesterol (mg/g WW)	Triglycerides (mg/g WW)	Liver Fat (%)	GSH (µmol/g WW)
NC	4.73 ± 2.47 ^b^	4.21 ± 3.82 ^b^	7.85 ± 1.20 ^ab^	11.87 ± 1.26 ^a^
HFD	12.13 ± 1.85 ^a^	13.06 ± 4.92 ^a^	9.98 ± 3.99 ^a^	6.43 ± 1.42 ^b^
HFD + ME	7.80 ± 3.85 ^ab^	6.14 ± 0.32 ^ab^	4.07 ± 0.93 ^b^	11.42 ± 1.84 ^a^
HFD + PE	3.37 ± 1.26 ^b^	4.31 ± 2.45 ^b^	3.91 ± 1.23 ^b^	12.12 ± 0.62 ^a^

WW: wet weight; NC: negative control; HFD: high-fat diet; HFD + ME: HFD + mallow extract; HFD + PE: HFD + parsley extract. GSH: glutathione. Results expressed as mean ± SD; different superscripted letters (a, b) in the same column indicate a significant difference between groups’ means at *p* < 0.05.

**Table 8 foods-12-04303-t008:** The efficiency of boiled parsley and mallow leave extracts (200 mg/kg BW, oral administration) on kidney function (mg/dL) in HFD-fed rats, *n* = 6.

Groups	Creatinine	Urea
NC	0.26 ± 0.0 ^b^	55.01 ± 2.90 ^b^
HFD	0.41 ± 0.1 ^a^	65.76 ± 3.85 ^a^
HFD + ME	0.26 ± 0.0 ^b^	58.61 ± 1.72 ^b^
HFD + PE	0.29 ± 0.1 ^ab^	58.61 ± 2.16 ^b^

NC: negative control; HFD: high-fat diet; HFD + ME: HFD + mallow extract; HFD + PE: HFD + parsley extract. Results expressed as mean ± SD; different superscripted letters (a, b) in the same column indicate a significant difference between groups’ means at *p* < 0.05.

## Data Availability

Data are contained within the article.

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
