# Peer review of "Hypolipidemic, Hypoglycemic, and Ameliorative Effects of Boiled Parsley (Petroselinum crispum) and Mallow (Corchorus olitorius) Leaf Extracts in High-Fat Diet-Fed Rats"

_foods, 2023, doi:10.3390/foods12234303_

Round 1

Reviewer 1 Report

Comments and Suggestions for Authors

The authors provided original and interesting research. Its relevance stems from the great importance of the problem of obesity. The authors propose an approach based on the use of very simple and accessible tools and techniques. The work has primarily practical value, but was done at a high scientific level. The study is well presented and discussed. I only have a few minor comments:

1. Line 98, please provide the AOAC transcript.

2. Line 193, please check 9.8±52?

3. Line 233. There are two tables 1 in the text, please check the numbering of the tables

4. The authors purchased parsley and mallow at the local market. It seems to me that a few words need to be spent on how the identification of the plants was carried out. Maybe a qualified botanist confirmed it or something.

Author Response

The authors provided original and interesting research. Its relevance stems from the great importance of the problem of obesity. The authors propose an approach based on the use of very simple and accessible tools and techniques. The work has primarily practical value, but was done at a high scientific level. The study is well presented and discussed. I only have a few minor comments:

>> Thank you for your positive feedback, greatly appreciate your suggestions and notes, and we have duly addressed them.

  1. Line 98, please provide the AOAC transcript.

>> The transcript has been added.

  1. Line 193, please check 9.8±52?

>> The correct value is 9.8±0.52. The standard deviation value might be misplaced during copying from the original table. Thank you for bringing this to our attention.

  1. Line 233. There are two tables 1 in the text, please check the numbering of the tables

>> The tables have been correctly numbered.

  1. The authors purchased parsley and mallow at the local market. It seems to me that a few words need to be spent on how the identification of the plants was carried out. Maybe a qualified botanist confirmed it or something.

>> Thank you for the suggestion. Although both plants are well-known and easily distinguishable, a concise statement based on previous botanical descriptions, has been included in the “Materials” section, subsection “ingredients”, to clearly distinguish the two plants.

Reviewer 2 Report

Comments and Suggestions for Authors

The biological results presented in the manuscript are interesting and valuable; however, there are two main drawbacks to this study:

1. Lack of complete chemical analysis of the extracts obtained (at least HPLC analysis of the main compounds); In fact, we do not know what substances are responsible for biological activity - whether they are present in all cultivated varieties of these vegetables and in what concentrations. Would the use of heat treatment for other cultivars give the same results? Is it known which cultivars were tested? 

2. lack of positive controls in the bioassays. Appropriate drugs would have to be used.

The methodological part is not well described at every point.

Author Response

The biological results presented in the manuscript are interesting and valuable; however, there are two main drawbacks to this study:

  1. Lack of complete chemical analysis of the extracts obtained (at least HPLC analysis of the main compounds); In fact, we do not know what substances are responsible for biological activity - whether they are present in all cultivated varieties of these vegetables and in what concentrations. Would the use of heat treatment for other cultivars give the same results? Is it known which cultivars were tested? 

>> We understand your concerns regarding the absence of HPLC analysis for identifying the main compounds in the extracts. While HPLC analysis is undoubtedly a valuable technique for determining the composition and concentrations of specific substances, our study primarily focused on determining the total phenolic compounds, which might be sufficient to achieve the objectives we have set. Although HPLC analysis would indeed level up the findings, the primary goal was to investigate the overall impact of the treatment applied in major changes in the selected plants.  
Regarding the use of heat treatment on other cultivars, it is an interesting point to consider. While our study might not have specifically investigated the effects of heat treatment on different cultivars, it could be an avenue for future research to explore the available cultivars of parsley and mallow that are native to Saudi Arabia. We appreciate your valuable remarks, and we will take them into consideration for future investigations. As for the cultivars tested in our study, we focused on specific cultivars that are commonly found in the local markets. The two cultivars used for both parsley and mallow were clearly stated in the “Materials” section, under the sub-section “Ingredients”.

  1. lack of positive controls in the bioassays. Appropriate drugs would have to be used.

>> The positive group was fed a high-fat diet throughout the entire experiment, and the treatment groups were fed as well. Therefore, this approach effectively establishes a suitable positive group for studying obesity. To ensure clarity on this aspect, we have added a statement (lines 161-162) to emphasize the feeding protocol. Although including a comparative drug administration to create a comparative positive group would enhance the study, obesity studies can still be adequately conducted without it.

The methodological part is not well described at every point.

>> Appreciate your note. Major enhancements have been made within the methodological section at multiple points, with particular emphasis on the biochemical analysis.

Reviewer 3 Report

Comments and Suggestions for Authors

The article shows interesting work on the emergence of heat treatments used in industry and how these can improve their biological activity. Major changes should be made to improve the quality of the manuscript:

The abstract is too long and exceeds the word limit for this journal. The order of the abract has not been followed, as this section has to be divided into introduction, hypothesis, materials, etc. (see instructions).

In this section the following is written backwards "boiled parsley and mallow leaf aqueous extracts (ME and PE, re-20 spectively)" with ME referring to mallow and PE to parsley.

Line 65 and 117, put phenolic instead of phenol.

Line 103. Specify the other component of methanol, if water, put methanol:water (7:3, v/v). Same on line 110 with sodium carbonate or in line 170 wit formalin solution.

Line 106 and 151: centripetal force should be expressed in g not rpm.

Results: The values in all tables may not be repeated in the text, unless you want to emphasise a particular value, but not all of them.

Table 1. TDS should be described in the table footnote, as each table or figure should be self-explanatory and not rely on acronyms described in the text.

Table 2, DPPH should be changed to antioxidant capacity which is how the values are actually expressed.

Some factors characterised as ash change significantly from one treatment to another and yet have not been discussed in the discussion.

Line 495. It is claimed that more complex phenols are broken down by heat treatments increasing total phenol content and antioxidant activity. But in reality, hydrolysis of the more complex phenols should not increase the quantification of total phenols, although it may increase antioxidant activity.

Regarding the discussion on the characterised values for the two substrates in the three types of treatments, the authors limit themselves to comparing the results with other bibliographic results without discussing why some data increase or decrease. Therefore, the discussion should be improved by providing explanations on the behaviour of the components with respect to the treatments applied.

The animal intervention study shows important differences with respect to the extracts obtained, but it would be of great interest to be able to correlate these changes with the concentration of the main components such as phenolic compounds or fibre by calculating their correlation coefficients.

Author Response

The article shows interesting work on the emergence of heat treatments used in industry and how these can improve their biological activity. Major changes should be made to improve the quality of the manuscript:

The abstract is too long and exceeds the word limit for this journal. The order of the abract has not been followed, as this section has to be divided into introduction, hypothesis, materials, etc. (see instructions).

>> Appreciate your valuable note. The abstract has been rewritten, properly formatted, and shortened from 298 to 208 words.

In this section the following is written backwards "boiled parsley and mallow leaf aqueous extracts (ME and PE, respectively)" with ME referring to mallow and PE to parsley.

>> Indeed you’re right, this has been fixed. Thank you.

Line 65 and 117, put phenolic instead of phenol.

>> Mentioned mistake has been corrected, appreciate it.

Line 103. Specify the other component of methanol, if water, put methanol:water (7:3, v/v). Same on line 110 with sodium carbonate or in line 170 wit formalin solution.

>> The mentioned statements have been corrected.

Line 106 and 151: centripetal force should be expressed in g not rpm.

>> The mentioned mistake has been corrected.

Results: The values in all tables may not be repeated in the text, unless you want to emphasise a particular value, but not all of them.

>> Thank you for the note. The repeated values in the results paragraphs have been minimized or replaced with their respective percentage equivalents to enhance the clarity of the text.

Table 1. TDS should be described in the table footnote, as each table or figure should be self-explanatory and not rely on acronyms described in the text.

>> Indeed, TDS was previously described in the footnote of Table 1.

Table 2, DPPH should be changed to antioxidant capacity which is how the values are actually expressed.

>> This has been correctly fixed.

Some factors characterised as ash change significantly from one treatment to another and yet have not been discussed in the discussion.

>> More emphasis on the changes in ash content has been provided in the discussion section.

Line 495. It is claimed that more complex phenols are broken down by heat treatments increasing total phenol content and antioxidant activity. But in reality, hydrolysis of the more complex phenols should not increase the quantification of total phenols, although it may increase antioxidant activity.

>> Indeed your point of view is evident, we acknowledge that other potential factors may contribute to the observed increase in the quantification of total phenols. These factors have been thoroughly explained in the discussion section, lines 558‒572.

Regarding the discussion on the characterised values for the two substrates in the three types of treatments, the authors limit themselves to comparing the results with other bibliographic results without discussing why some data increase or decrease. Therefore, the discussion should be improved by providing explanations on the behaviour of the components with respect to the treatments applied.

>> Major enhancements have been made within the discussion section to emphasize and explain the observed findings; in-depth analysis and explanations have been provided in lines 485‒494, 509‒522, and 558‒572. Appreciate your valuable note.

The animal intervention study shows important differences with respect to the extracts obtained, but it would be of great interest to be able to correlate these changes with the concentration of the main components such as phenolic compounds or fibre by calculating their correlation coefficients.

>> Greatly appreciate your valuable and interesting suggestion. Calculating the correlation coefficients would indeed add great value to the obtained findings. However, it is important to note and emphasize that the extract used in this study is not intended for use as an isolated supplement or in drug capsules. The aim of the study was to highlight and investigate the influence of boiling treatment, among the other treatments, on the anti-obesity potential of the tested plants. It should be also noted that we indeed highlighted the potential associations between the observed findings and the changes observed in phenolic compounds and fiber content, for instance.

Round 2

Reviewer 2 Report

Comments and Suggestions for Authors

The actual manuscript is not attached. The manuscript "Insects as Valuable Sources of Protein and Peptides..." opens.

However, if the authors made corrections in the methodological part and specified the plant varieties used then I think the manuscript can be published. 

Author Response

Major corrections and improvements have indeed been made in the methodological part. It appears that a technical issue may have occurred, potentially resulting in the mix-up of a different manuscript. Nonetheless, a new manuscript submission will be uploaded during this round of revision. We would appreciate any additional suggestions or feedback that you may have.

Reviewer 3 Report

Comments and Suggestions for Authors

In the results, many values that are presented in the tables are still repeated in the text and should therefore not be repeated.

In lines 561 to 564, it is stated that chemical hydrolysis of certain tannins increases the total phenolic content. This statement is not completely true since the method for determining total phenolics does not depend on the degree of hydrolysis of these tannins but on their functional groups, so the content should be the same with or without hydrolysis. What we proposed as the most correct and proven theory in other plant materials to the authors is that the chemical hydrolysis helps to free the phenolics that bind to the cell wall material and by freeing them do give the colour reaction for the determination of total phenolics.

Author Response

In the results, many values that are presented in the tables are still repeated in the text and should therefore not be repeated.

>> The repeated values in the text were excluded or replaced with their percentage difference equivalent. These edits were made using track changes and the font color was changed to red. Appreciate your note, it has indeed improved the result's clarity.

In lines 561 to 564, it is stated that chemical hydrolysis of certain tannins increases the total phenolic content. This statement is not completely true since the method for determining total phenolics does not depend on the degree of hydrolysis of these tannins but on their functional groups, so the content should be the same with or without hydrolysis. What we proposed as the most correct and proven theory in other plant materials to the authors is that the chemical hydrolysis helps to free the phenolics that bind to the cell wall material and by freeing them do give the colour reaction for the determination of total phenolics.

>> Appreciate your feedback. Thank you for pointing out the importance of considering the functional groups of tannins rather than their degree of hydrolysis in determining total phenolic content. We acknowledge that the theory you mentioned as the most accurate and well-established one, indeed has already been stated earlier in the discussion.

Our intention was to highlight that in certain situations, tannins might facilitate the release of phenolic compounds during processes such as boiling and blanching due to their ability to form complexes with phenolics, and when exposed to heat, these complexes may break down, leading to the release of phenolics into the surrounding liquid or cooking medium. This can potentially increase the bioavailability of phenolics. However, based on your explanation and point of view, the statement in lines 561 to 564 has been excluded to limit any misunderstanding and emphasize the most acknowledged and proven theory.

Thank you once again for your valuable input.